# OpenReview forum: "Human-Guided Harm Recovery for Computer Use Agents"
_ICLR.cc/2026/Conference — ICLR 2026 Conference Withdrawn Submission_

### Official Review · Reviewer_yNia · 2025-10-29

**Soundness:** 3
**Presentation:** 3
**Contribution:** 3
**Rating:** 2
**Confidence:** 3

**Summary:**

This paper presents a method for detecting and recovering from harmful behaviors in computer use agents. Specifically, it proposes a framework to transform an agent from a harmful state to a safe state, in addition to optimally identifying the most effective remedy for the harm. The proposed system, Agent Scaffold, employs two language models: one dedicated to generating recovery plans and another to verifying and evaluating these plans. For evaluation, the authors introduce Backbench, a benchmark comprising 50 backtracking scenarios designed to test Agent Scaffold to generate recovery plans and to select those most preferred by the human rubric for returning agents from harmful to safe states.

**Strengths:**

1-	The proposed method not limited to harm prevention, it also focuses on recovering agents from harmful states.

2-	The proposed approach integrates human judgment (Human Preference Rubric) in selecting the most effective recovery plan.

3-	The paper introduces new benchmark (Backbench) containing 50 diverse backtracking scenarios across multiple categories.

**Weaknesses:**

1-	The overall flow of the paper is difficult to follow, especially the description of the harm detection and recovery process. It would be helpful to present the sequence of steps starting from harm occurrence and detection to recovery execution.

2-	The harm detection process is not well explained. It is unclear whether the paper assumes harm is already detected and focuses solely on the recovery phase or the proposed method will detect the harm as well as introduce and execute the plan of recovery.

3-	The process of comparing recovery plans lacks details regarding computational cost because many plans can be generated by LMgen. Is LMgen limited to specific number of plans to generate? and what is the estimate time for LMver to compare and select from these plans?

4-	The rubric extraction procedure in Section 4.1 is unclear. The paper states that 20 human judges were involved, but it does not explain how potential biases were mitigated or how evaluation consistency was ensured.

5-	The experimental results is limited to the Backbench dataset. The effectiveness of Agent Scaffold on additional public benchmarks for agent safety, such as OS-HARM, was not shown.

6-	On the evaluation section, one of the comparing factors is agent with no plan. This factor is not clear, in other words, does agent with no plan have pre-execution safeguards? If not, it would be helpful to see the results/performance for agents with pre-execution safeguards on the proposed benchmark.

7-	Figure 5 in the results section shows that the human rubric and the single-plan approach are consistently close to each other. More explanation regarding these results is needed. If the cost of comparing different plans is high, wouldn’t it be sufficient to use the single plan generated by LMgen to return the agent to a safe state?

**Questions:**

Please refer to the weaknesses points 3- 7.

---

### Official Review · Reviewer_cm1S · 2025-10-30

**Soundness:** 3
**Presentation:** 2
**Contribution:** 2
**Rating:** 2
**Confidence:** 4

**Summary:**

The paper aims to study the ability of Computer Use Agents to fix states of harm introduced by agent's actions. First, it collects BackBench, a dataset of scenarios with different types of harms in the OSWorld environment. Then, it derives a human-informed rubric from human preference data to help an LLM verifier to compare the quality of possible remedy agent plans. At the test-time an LLM generator proposes a set of plans for the agent to solve the task, and the verifier needs to identify the one which best represents human preferences. In the evaluation on BackBench, the solutions suggested by the proposed approach are preferred to those of the baselines by both human and automated judges.

**Strengths:**

- The paper takes a different angle about evaluating the safety capabilities of CUA, i.e. fixing existing issues (created by the agent itself) rather than preventing unsafe actions. BackBench is introduced for this goal.

- The proposed scaffolding and plan selection is reasonable, and can effectively identify solutions preferred by the judges.

**Weaknesses:**

- The agent doesn't need to detect the existing safety threat, but it's simply given the task of fixing it. This is not necessarily realistic (a different harm classifier would be needed), and seems to boil down to evaluating the generic CUA capabilities in a narrow task.

- In the evaluation, the proposed scaffolding gets better results than the baselines. However, in App. A.2.3, in the instructions on how to choose the better trajectory, a guideline clearly states to prefer faster solutions of the task, which is also one of the criteria prioritized in the rubric. Moreover, the paper doesn't report the actual success rate of the agents, i.e. how often the issue is actually fixed.

- The preferences emerged by the rubric distillation seem quite straightforward (good and efficient solutions are preferred), so it's not clear which additional insights are provided.

**Questions:**

Overall, the proposed approach consists in tuning a prompt to choose which are the plans preferred by humans to solve a narrow task. However, the improvements in term of success rate in solving the task and how this framework is integrated in a general CUA are not clear.

---

### Official Review · Reviewer_h3Aq · 2025-11-02

**Soundness:** 2
**Presentation:** 2
**Contribution:** 3
**Rating:** 4
**Confidence:** 3

**Summary:**

This study investigates a novel research problem, post-execution harm recovery. It constructs BackBench, a benchmark of 50 computer-use tasks that evaluates the ability of CUAs to recover from harmful states. The results show current CUAs perform poorly on mitigating and backtracking from states of harm. To tackle the harm recovery problem, the authors conduct a formative user study and contribute a preference dataset of human judgments on recovery plans. Based on this annotation dataset, they extract a principled rubric of recovery plan attributes and then propose a new generate-and-verify scaffold that incorporates this rubric. The experiment results show that the scaffold improves agent backtrack performance under both human and LLM evaluation.

**Strengths:**

- The authors conduct a formative user study to help define how human users judge recovery quality, which ends up with a natural language rubric and a dataset of human annotations on preference judgment of recovery plans.
- This study introduces BackBench, a novel benchmark consisting of 50 backtrack scenarios that evaluates the CUA's capability of recovering from harmful states.
- Moreover, it proposes a novel agent scaffold to improve the agent's recovery ability by incorporate human preference at test time.
- Provide a method to approximate the solution of intractable reward optimization problem in the absence of execution trajectories, using a generate-and-very scaffold augmented by human preferences.

**Weaknesses:**

- The authors should discuss about the implication of low inter-annotator agreement of human preference rates.
- No evaluation on the reliability and feasibility of the LM-based verifier.
- I doubt the feasibility of prompting LLMs like Claude-4-sonnet to generate candidate recovery plans and distinguish the effective plan from ineffective ones, given the fact that the model itself fails to recover from harmful states. Please comment on this problem.
- The authors only validate the proposed scaffold on claude-4-sonnet. It'd be necessary to run evaluations on more models, including both open-sourced and proprietary models.

**Questions:**

- Did you collect human annotations on the recovery plans from AI agent safety experts? I believe that the expert annotations are more representative and significant for this harm recovery problem since normal computer users are often not aware of how many security loopholes they have. Different from normal question-answering tasks, the recovery plan that is favored by most of people may be not objectively correct.

---

### Official Review · Reviewer_tv2K · 2025-11-02

**Soundness:** 2
**Presentation:** 2
**Contribution:** 2
**Rating:** 2
**Confidence:** 4

**Summary:**

This paper introduces a novel paradigm for AI agent safety focused on post-execution harm recovery rather than solely pre-execution prevention. The authors formalize harm recovery as an optimization problem over human preferences, develop BackBench—a benchmark of 50 computer-use scenarios requiring agents to recover from harmful states—and propose a generate-and-verify scaffold that uses human preference signals to select recovery plans. Through user studies with 250 participants, they derive a rubric of 8 recovery attributes and collect 1,150 pairwise preference judgments. Their human-preference guided approach significantly outperforms baselines under both human and LLM evaluation.

**Strengths:**

- I like the fresh take on switching attention to post-execution recovery instead of pre-execution prevention. The agentic setting is very open-ended, so it makes sense that the current literature on agent safety might have overfocused on prevention, while the actual action space for defenders is significantly larger.
- The benchmark is built on top of OSWorld which seems like the right choice. OSWorld represents an environment where backtracking is non-trivial to implement (unlike, e.g., coding environments where often it’s easy to revert back to a previous commit).

**Weaknesses:**

- Only one agent backbone is used (`claude-4-sonnet-20250514`), while it should be easy to evaluate multiple backbones of different sizes and from different providers (e.g., [OS-Harm](https://arxiv.org/abs/2506.14866) tests `o4-mini`, `gpt-4.1`, `gemini-2.5-pro`, `gemini-2.5-flash` in addition to a Sonnet model). Having only one agent backbone provides an incomplete picture.
- It would be great to have a larger number of computer use tasks (i.e., more than 50).
- *“In total, we collected a dataset of 1150 annotated plan pairs, with 150 pairs independently rated by two annotators to measure inter-annotator agreement; 230 total annotators participated in the ratings. Inter-annotator agreement was quantified using Cohen’s κ, which yielded a value of 0.15, indicating relatively low agreement under the conventional interpretation of this statistic.”* - The inter-annotator agreement is very low, so it’s not clear how useful this dataset is.
- Using GPT-4.1 to judge recovery quality when the verifier also uses LLMs creates potential circularity. While human evaluation validates results to some extent (although human inter-annotator agreement is fairly low), the correlation between human and LLM judgments deserves deeper analysis.

Overall, the benchmark does seem interesting, but I would say it’s still in a bit of a preliminary stage.

**Questions:**

None.

---

### Note · Authors · 2025-11-20

**Comment:**

We would like to thank the reviewers for their feedback and constructive suggestions. We plan to incorporate their comments into a revised version of the paper for a future submission. Thank you again for your time and effort in evaluating our work.

**Withdrawal Confirmation:**

I have read and agree with the venue's withdrawal policy on behalf of myself and my co-authors.